# Neural Architecture Search
# without Training

## Abstract

The time and effort involved in hand-designing deep neural networks is immense. This has prompted the development of Neural Architecture Search (NAS) techniques to automate this design. However, NAS algorithms tend to be slow and expensive; they need to train vast numbers of candidate networks to inform the search process. This could be remedied if we could infer a network's trained accuracy from its initial state. In this work, we examine the correlation of linear maps induced by augmented versions of a single image in *untrained* networks and motivate how this can be used to give a measure which is highly indicative of a network's *trained* performance. We incorporate this measure into a simple algorithm that allows us to search for powerful networks without any training in a matter of seconds on a single GPU, and verify its effectiveness on NAS-Bench-101 and NAS-Bench-201. Finally, we show that our approach can be readily combined with more expensive search methods for added value: we modify regularised evolutionary search to produce a novel algorithm that outperforms its predecessor.

## 1 Introduction

The success of deep learning in computer vision is in no small part due to the insight and engineering efforts of human experts, allowing for the creation of powerful architectures for widespread adoption (Krizhevsky et al., 2012; Simonyan & Zisserman, 2015; He et al., 2016; Szegedy et al., 2016; Huang et al., 2017). However, this manual design is costly, and becomes increasingly more difficult as networks get larger and more complicated. Because of these challenges, the neural network community has seen a shift from designing architectures to designing algorithms that *search* for candidate architectures (Elsken et al., 2019; Wistuba et al., 2019). These Neural Architecture Search (NAS) algorithms are capable of automating the discovery of effective architectures (Zoph & Le, 2017; Zoph et al., 2018; Pham et al., 2018; Tan et al., 2019; Liu et al., 2019; Real et al., 2019).

NAS algorithms are broadly based on the seminal work of Zoph & Le (2017). A controller network generates an architecture proposal, which is then trained to provide a signal to the controller through REINFORCE (Williams, 1992), which then produces a new proposal, and so on. Training a network for every controller update is extremely expensive; utilising 800 GPUs for 28 days in Zoph & Le (2017). Subsequent work has sought to ameliorate this by (i) learning stackable cells instead of whole networks (Zoph et al., 2018) and (ii) incorporating *weight sharing*; allowing candidate networks to share weights to allow for joint training (Pham et al., 2018). These contributions have accelerated the speed of NAS algorithms e.g. to half a day on a single GPU in Pham et al. (2018).

For some practitioners, NAS is still too slow; being able to perform NAS quickly (i.e. in seconds) would be immensely useful in the hardware-aware setting where a separate search is typically required for each device and task (Wu et al., 2019; Tan et al., 2019). Moreover, recent works have scrutinised NAS with weight sharing (Li & Talwalkar, 2019; Yu et al., 2020); there is continued debate as to whether it is clearly better than simple random search.

The issues of cost and time, and the risks of weight sharing could be avoided entirely if a NAS algorithm *did not require any network training*. In this paper, we show that this can be achieved. We explore two recently released NAS benchmarks, NAS-Bench-101 (Ying et al., 2019), and NAS-Bench-201 (Dong & Yang, 2020) and examine the relationship between the linear maps induced by an **untrained** network for a minibatch of augmented versions of a *single image* (Section 3). These maps are easily computed using the Jacobian. The correlations between these maps (which we denote

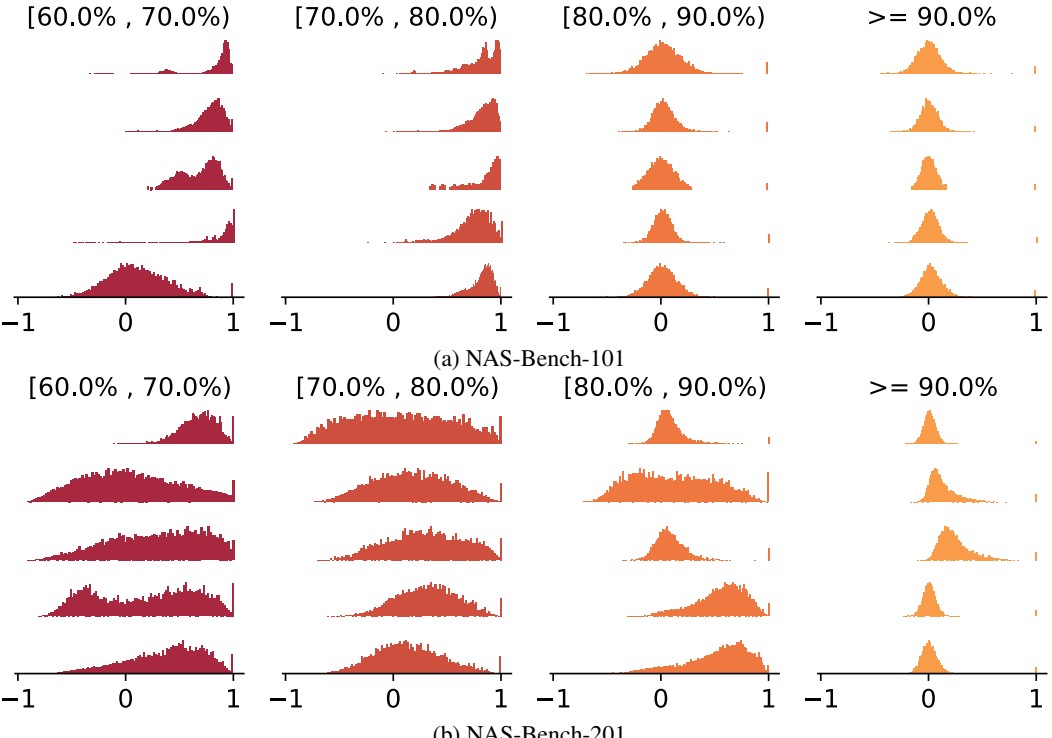

Figure 1: Histograms of the correlations between linear maps in an augmented minibatch of a single CIFAR-10 image for **untrained** architectures in (a) NAS-Bench-101 (b) NAS-Bench-201. The histograms are sorted into columns based on the final CIFAR-10 validation accuracy **when trained**. The y-axes are individually scaled for visibility. The profiles are distinctive; the histograms for good architectures in both search spaces have their mass around zero with a small positive skew. We can look at this distribution for an untrained network to predict its final performance without any training. More histograms are available in Appendix A.

by $\Sigma_J$) are distinctive for networks that perform well when trained on both NAS-Benches; this is immediately apparent from visualisation alone (Figure 1). We devise a score based on $\Sigma_J$ and perform an ablation study to demonstrate its robustness to inputs and network initialisation.

We incorporate our score into a simple search algorithm that **doesn't require training** (Section 4). This allows us to perform architecture search quickly, for example, on CIFAR-10 (Krizhevsky, 2009) we are able to search for a network that achieve 93.36% accuracy in 29 seconds within the NAS-Bench-201 search space; several orders of magnitude faster than traditional NAS methods for a modest change in final accuracy (e.g. REINFORCE finds a 93.85% net in 12000 seconds). Finally, we show that we can combine our approach with regularised evolutionary search (REA, Pham et al., 2018) to produce a new NAS algorithm, *Assisted-REA* (AREA) that outperforms its precedessor, attaining 94.16% accuracy on NAS-Bench-101 in 12,000 seconds. Code for reproducing our experiments is available in the supplementary material.

We believe this work is an important proof-of-concept for NAS without training, and shows that the large resource costs associated with NAS can be avoided. The benefit is two-fold, as we also show that we can integrate our approach into existing NAS techniques for scenarios where obtaining as high an accuracy as possible is of the essence.

## 2 BACKGROUND

Designing a neural architecture by hand is a challenging and time-consuming task. It is extremely difficult to intuit where to place connections, or which operations to use. This has prompted an abundance of research into neural architecture search (NAS); the automation of the network design process. In the pioneering work of Zoph & Le (2017), the authors use an RNN controller to generate

descriptions of candidate networks. Candidate networks are trained, and used to update the controller using reinforcement learning to improve the quality of the candidates it generates. This algorithm is very expensive: searching for an architecture to classify CIFAR-10 required 800 GPUs for 28 days. It is also inflexible; the final network obtained is fixed and cannot be scaled e.g. for use on mobile devices or for other datasets.

The subsequent work of Zoph et al. (2018) deals with these limitations. Inspired by the modular nature of successful hand-designed networks (Simonyan & Zisserman, 2015; He et al., 2016; Huang et al., 2017), they propose searching over neural building blocks, instead of over whole architectures. These building blocks, or *cells*, form part of a fixed overall network structure. Specifically, the authors learn a standard cell, and a reduced cell (incorporating pooling) for CIFAR-10 classification. These are then used as the building blocks of a larger network for ImageNet (Russakovsky et al., 2015) classification. While more flexible—the number of cells can be adjusted according to budget— and cheaper, owing to a smaller search space, this technique still utilises 500 GPUs across 4 days.

ENAS (Pham et al., 2018) reduces the computational cost of searching by allowing multiple candidate architectures to share weights. This facilitates the simultaneous training of candidates, reducing the search time on CIFAR-10 to half a day on a single GPU. Weight sharing has seen widespread adoption in a host of NAS algorithms (Liu et al., 2019; Luo et al., 2018; Cai et al., 2019; Xie et al., 2019; Brock et al., 2018). However, there is evidence that it inhibits the search for optimal architectures (Yu et al., 2020). Moreover, random search proves to be an extremely effective NAS baseline (Yu et al., 2020; Li & Talwalkar, 2019). This exposes another problem: the search space is still vast—there are $1.6 \times 10^{29}$ possible architectures in Pham et al. (2018) for example—that it is impossible to isolate the best networks and demonstrate that NAS algorithms find them.

An orthogonal direction for identifying good architectures is the estimation of accuracy prior to training (Deng et al., 2017; Istrate et al., 2019), although these differ from this work in that they rely on training a predictive model, rather than investigating more fundamental architectural properties.

## 2.1 NAS BENCHMARKS

A major barrier to evaluating the effectiveness of a NAS algorithm is that the search space (the set of all possible networks) is too large for exhaustive evaluation. Moreover, popular search spaces have been shown to be over-engineered, exhibiting little variety in their trained networks (Yang et al., 2020). This has led to the creation of several benchmarks (Ying et al., 2019; Zela et al., 2020; Dong & Yang, 2020) that consist of tractable NAS search spaces, and metadata for the training of networks within that search space. Concretely, this means that it is now possible to determine whether an algorithm is able to search for a good network. In this work we utilise NAS-Bench-101 (Ying et al., 2019) and NAS-Bench-201 (Dong & Yang, 2020) to evaluate the effectiveness of our approach. NAS-Bench-101 consists of 423,624 neural networks that have been trained exhaustively, with three different initialisations, on the CIFAR-10 dataset for 108 epochs. NAS-Bench-201 consists of 15,625 networks trained multiple times on CIFAR-10, CIFAR-100, and ImageNet-16-120 (Chrabaszcz et al., 2017). Both benchmarks are described in detail in Appendix B.

## 3 SCORING NETWORKS AT INITIALISATION

Our goal is to devise a means to score a network architecture at initialisation in a way that is indicative of its final trained accuracy. This can either replace the expensive inner-loop training step in NAS, or better direct exploration in existing NAS algorithms.

Given a neural network with rectified linear units, we can, at each unit in each layer, identify a binary indicator as to whether the unit is inactive (the value is negative and hence is multiplied by zero) or active (in which case its value is multiplied by one). Fixing these indicator variables, it is well known that the network is now locally defined by a linear operator (Hanin & Rolnick, 2019); this operator is obtained by multiplying the linear maps at each layer interspersed with the binary rectification units. Consider a minibatch of data $\mathbf{X} = \{\mathbf{x}_i\}_{i=1}^N$. Let us denote the linear map for input $\mathbf{x}_i \in \mathbb{R}^D$ by column vector $\mathbf{w}_i$, which maps the input through the network $f(\mathbf{x}_i)$ to a final choice of scalar representation $z_i \in \mathbb{R}^1$. This linear map can be easily computed using the Jacobian $\mathbf{w}_i = \frac{\partial f(\mathbf{x}_i)}{\partial \mathbf{x}}$.

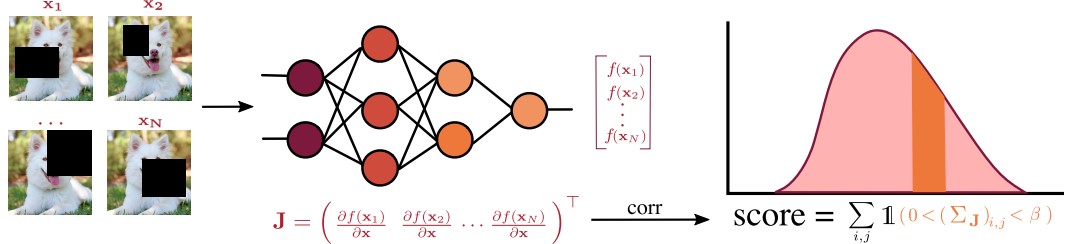

Figure 2: How to score **untrained** networks: (i) Create a minibatch of an image repeatedly augmented with cutout; (ii) Perform a forward and backward pass using that minibatch to compute $\mathbf{J}$; (iii) Compute the correlation matrix for $\mathbf{J}$ and count the number of entries in it that lie between 0 and $\beta$.

How differently a network acts at each data point can be summarised by comparing the corresponding local linear operators. Correlated operators for nearby points (such as small perturbations from a training point) relate to a potential difficulty in handling the two points differently during learning. The Frobenius inner product $\mathrm{Tr}[(\mathbf{w}_i - \mu_i)^T(\mathbf{w}_j - \mu_j)]$ provides a natural basis for defining how two linear operators corresponding to data points $\mathbf{x}_i$ and $\mathbf{x}_j$ covary ($\mu$ are mean Jacobian elements, and usually close to zero). We can examine the correspondences for the whole minibatch by computing

$$\mathbf{J} = \left( \frac{\partial f(\mathbf{x}_1)}{\partial \mathbf{x}} \quad \frac{\partial f(\mathbf{x}_2)}{\partial \mathbf{x}} \quad \cdots \quad \frac{\partial f(\mathbf{x}_N)}{\partial \mathbf{x}} \right)^{\top} \tag{1}$$

and observing the covariance matrix $\mathbf{C}_J = (\mathbf{J} - \mathbf{M}_J)(\mathbf{J} - \mathbf{M}_J)^T$ where $\mathbf{M}_J$ is the matrix with entries $(\mathbf{M}_J)_{i,t} = \frac{1}{D}\sum_{d=1}^{D}\mathbf{J}_{i,d} \; \forall t$, where $d, t$ index over the $D$ elements of each input (i.e. channels $\times$ pixels). It is more salient to focus on the the correlation matrix $\mathbf{\Sigma}_J$ as the appropriate scaling in input space around each point is arbitrary. The $(i, j)^{th}$ element of $\mathbf{\Sigma}_J$ is given by $(\mathbf{\Sigma}_J)_{i,j} = \frac{(\mathbf{C}_J)_{i,j}}{\sqrt{(\mathbf{C}_J)_{i,i}(\mathbf{C}_J)_{j,j}}}$.

We want an untrained neural network to be sufficiently flexible to model a complex target function. However, we also want a network to be invariant to small perturbations. These two requirements are antagonistic. For an untrained neural network to be sufficiently flexible it would need to be able to distinguish the local linear operators associated with each data point: if two are the same then the two points are coupled. To be invariant to small perturbations the same local linear operators would need to be weakly coupled. Ideally a network would have low correlated local maps associated with each data point to be able to model each local region.

We empirically demonstrate this by computing $\mathbf{\Sigma}_J$ for a random subset of NAS-Bench-101 (Ying et al., 2019) and NAS-Bench-201 (Dong & Yang, 2020) networks **at initialisation** for a minibatch of a single CIFAR-10 image replicated 256 times with a different cutout (DeVries & Taylor, 2017) perturbation applied to each replicant. We use `torchvision.transforms.RandomErasing(p=0.9, scale=(0.02, 0.04))` in our experiments. To form $\mathbf{J}$ we flatten the Jacobian for each input (so $D = 3 \times 32 \times 32 = 3072$), and adjust the final classifier layer to output a scalar. The plots of the histograms of $\mathbf{\Sigma}_J$ for different networks, categorised according to the validation accuracy **when trained** is given in Figure 1 for a sample of networks in both benchmarks. Further plots are given in Appendix A.

The histograms are very distinct: high performing networks in both benchmarks have their mass tightly around zero with a small positive skew. We can therefore use these histograms to predict the final performance of untrained networks, in place of the expensive training step in NAS. Specifically, we score networks by counting the entries in $\mathbf{\Sigma}_J$ that lie between 0 and an small upper bound $\beta$. A $\mathbf{\Sigma}_J$ where inputs are marginally positively-correlated will have a higher score. Our score is given by

$$S = \sum_{i,j} \mathbb{1}\left(0 < (\mathbf{\Sigma_J})_{i,j} < \beta\right) \tag{2}$$

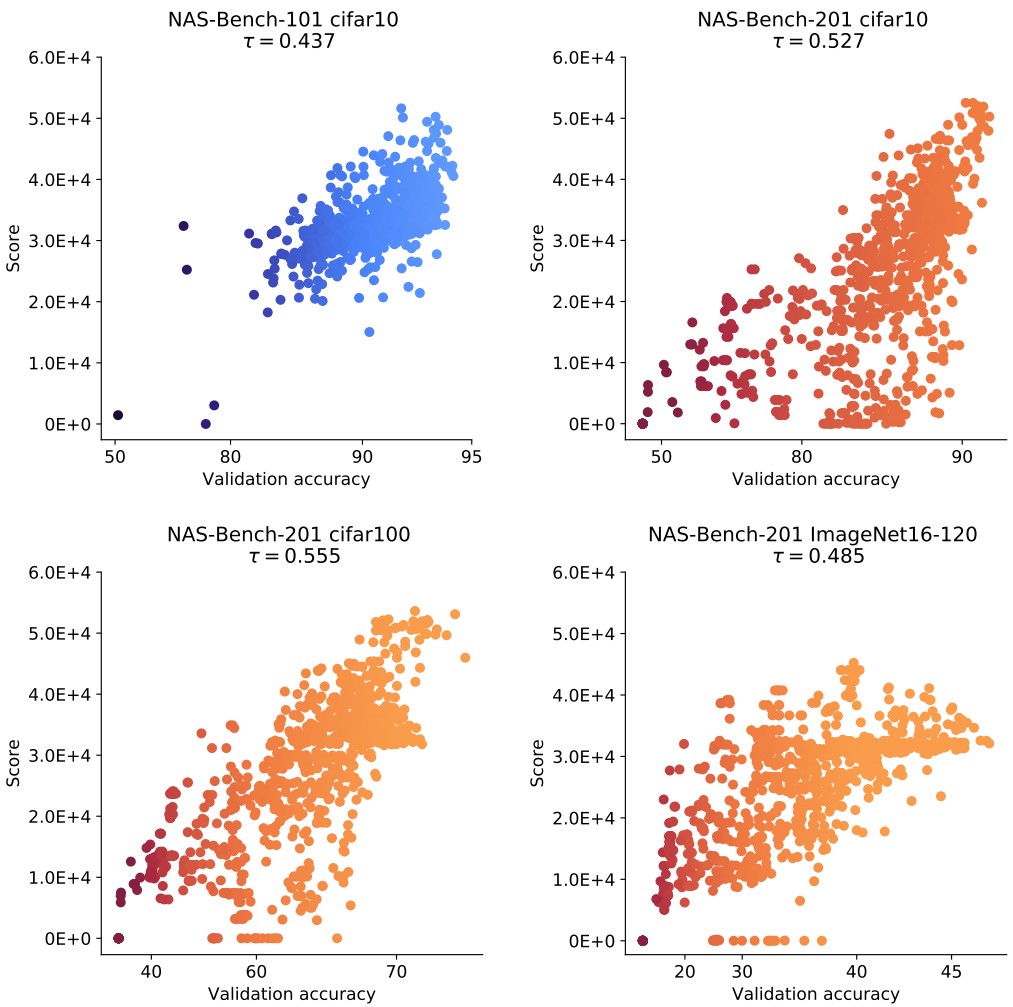

Figure 3: Plots of our score for 1000 randomly sampled **untrained** architectures in NAS-Bench-101 and NAS-Bench-201 against validation accuracy when trained. The inputs when computing the score and the validation accuracy for each plot are from CIFAR-10 (top), CIFAR-100 (bottom-left) and ImageNet16-120 (bottom-right). In all cases there is a noticeable correlation between the score for an untrained network and the final accuracy when trained.

where $\mathbb{1}$ is the indicator function. In this work we set $\beta = \frac{1}{4}$. An overview is provided in Figure 2. To be clear, we are not claiming this score is particularly optimal; rather we use it to demonstrate that there are ways of scoring untrained networks that provide significant value for architecture search.

We sample 1000 different architectures at random from NAS-Bench-101 and NAS-Bench-201 and plot our score on the *untrained* network versus their validation accuracies *when trained* for the datasets in these benchmarks in Figure 3. In all cases there is a strong correlation between our score and the final accuracy, although it is noisier for ImageNet-16-120; this dataset has smaller images compared to the other datasets so it may be that different cutout parameters would improve this. In Section 4 we demonstrate how our score can be used in a NAS algorithm for extremely fast search.

### 3.1 ABLATION STUDY

**How important is the image used to compute the score?** Our score relies on a single image being repeatedly augmented to form a minibatch. To determine how important the image is we randomly select 9 architectures in NAS-Bench-201 and compute the score separately for 10 random CIFAR-10 images. The resulting box-and-whisker plot is given in Figure 4(left): different architectures vary in

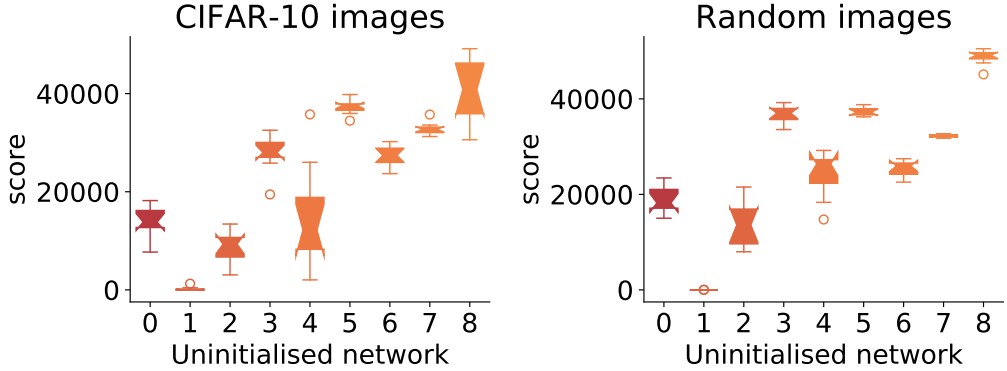

Figure 4: Effect of different CIFAR-10 image (left), and random input images (right) for 9 randomly selected NAS-Bench-201 architectures (one in each $10\%$ percentile range from 10-20, ..., 90-100). Random input data are normally distributed. For each network 10 samples were taken for each ablation. There is less variance in score when using different random input images as opposed to different CIFAR-10 images.

sensitivity, although the ranking of the scores is reasonably robust to the specific choice of image. In Figure 4(right) we compute our score using normally distributed random inputs, which seems to have little impact on the general trend. This leads us to believe the score captures a property of the network architecture, rather than something data-specific.

**Does the score change for different intialisations?** Figure 7(bottom-left) in Appendix C shows the score for different initial weights. This has a small effect on the score, supporting the idea that we are quantifying an architectural property.

**Does the size of the minibatch matter?** The size of $\Sigma_J$ scales with minibatch size. As our score sums entries of this matrix, we need to normalise our score to compare different minibatch sizes. Figure 7(bottom-right) in Appendix C presents this normalised score for different minibatch sizes, which shows little variation.

**What cell types score highly?** We were unable to detect any obvious patterns in the types of network architecture our score ranks highly. High scoring networks contained a range of operations, and connectivities. We leave a fine-grained analysis of this for future work.

## 4 NEURAL ARCHITECTURE SEARCH WITHOUT TRAINING - NASWOT

Table 1: Mean $\pm$ std. accuracy from NAS-Bench-101. NASWOT is our sampling algorithm (500 runs). REAL (our implementation of REA) uses evolutionary search to select an architecture (50 runs), Random selects one architecture (500 runs). AREAL (assisted-REAL) uses our score to select the starting population for REAL (50 runs). Search times for REAL and AREAL were calculated using the NAS-Bench-101 API.

| Method | Search (s) | CIFAR-10 |
|---|---|---|
| Random | N/A | 90.38±5.51 |
| NASWOT(N=100) | 55 | 92.95±0.88 |
| REAL | 12000 | 93.88±0.30 |
| AREAL (Ours) | 12000 | 94.16±0.18 |

In Section 3 we derived a score for cheaply ranking networks at initialisation based on their expected performance (Equation 2). Here as a proof of concept, we integrate this score into a simple search algorithm and evaluate its ability to alleviate the need for training in NAS. Code for reproducing our experiments is available in the supplementary material.

Many NAS algorithms are based on that of Zoph & Le (2017), which is illustrated in Algorithm 1. It consists of learning a generator network which proposes an architecture. The weights of the generator are learnt by training the networks it generates, either on a proxy task or on the dataset itself, and using their trained accuracies as signal through e.g. REINFORCE (Williams, 1992). This is repeated until the generator is

| **Algorithm 1** Standard NAS | **Algorithm 2** NASWOT |
|---|---|
| generator = GeneratorNet() | generator = RandomGenerator() |
| | best_net, best_score = None, 0 |
| **for** `i=1:N` **do** | **for** `i=1:N` **do** |
|     net = generator.generate() |     net = generator.generate() |
|     trained_net = net.train() |     score = net.score() |
|     ▷ *Training a net every step is expensive* |     **if** score > best_score **then** |
|     generator.update(trained_net) |         best_net, best_score = net, score |
| chosen_net = generator.generate() | chosen_net = best_network |

trained; it then produces a final network which is the output of this algorithm. The vast majority of the cost is incurred by having to train candidate architectures for every single controller update. Note that there exist alternative schema utilising e.g. evolutionary algorithms such as (Real et al., 2019) or bilevel optimisation (Liu et al., 2019) but all involve a training element.

We instead propose a simple alternative—NASWOT—illustrated in Algorithm 2. Instead of having a neural network as a generator, we randomly propose a candidate from the search space and then rather than training it, we score it in its untrained state using Equation 2. We do this `N` times—i.e. we have a sample size of `N` architectures—and then output the highest scoring network. As scoring only requires computing gradients for a single minibatch, this takes very little time.

**NAS-Bench-101.** We compare NASWOT to 12000 seconds of REAL—our implementation of REA (Real et al., 2019)—and random selection on NAS-Bench-101 (Ying et al., 2019) in Table 1. We can see that we are able to find a network with a final accuracy within a percent of REAL in under a minute on a single GPU.

**NAS-Bench-201.** Dong & Yang (2020) benchmarks a wide range of NAS algorithms, both with and without weight sharing, that we compare to NASWOT. The weight sharing methods are random search with parameter sharing (RSPS, Li & Talwalkar, 2019), first-order DARTS (DARTS-V1, Liu et al., 2019), second order DARTS (DARTS-V2, Liu et al., 2019), GDAS (Dong & Yang, 2019b), SETN (Dong & Yang, 2019a), and ENAS (Pham et al., 2018). The non-weight sharing methods are random search with training (RS), REA (Real et al., 2019), REINFORCE (Williams, 1992), and BOHB (Falkner et al., 2018). For implementation details we refer the reader to Dong & Yang (2020). The hyperparameters in NAS-Bench-201 for training and search are fixed — these results may not be invariant to hyperparameter choices, which may explain the low performance of e.g. DARTS.

We report results on the validation and test sets of CIFAR-10, CIFAR-100, and ImageNet-16-120 in Table 2. Search times are reported for CIFAR-10 on a single GeForce GTX 1080 Ti GPU. As per the NAS-Bench-201 setup, the non-weight sharing methods are given a time budget of 12000 seconds. For NASWOT and the non-weight sharing methods, accuracies are averaged over 500 runs. For weight-sharing methods, accuracies are reported over 3 runs. We report NASWOT for sample sizes of N=10 and N=100. With the exception of GDAS (Dong & Yang, 2019b), NASWOT is able to outperform all of the weight sharing methods while requiring a fraction of the search time.

The non-weight sharing methods do outperform NASWOT, though they also incur a large search time cost. REA for instance, also requires a large memory budget due to the maintenance of a population of parallel architectures which may suffer from scalability issues as model and dataset sizes increase. For CIFAR-10 and CIFAR-100 NASWOT is able to find networks with performance close to the best non-weight sharing algorithms, suggesting that network architectures themselves contain almost as much information about final performance at initialisation as after training.

Table 2 also shows the effect of sample size (N). We show the accuracy of networks chosen by our method for each N. We list optimal accuracy for each N, and random selection over the whole benchmark, both averaged over 500 runs. We observe that sample size increases NASWOT performance.

A key practical benefit of NASWOT is its execution time. Where time is a constraint, our method is very appealing. This may be important when repeating NAS several times, for instance for several hardware devices or datasets. This affords us the ability in future to specialise neural architectures for a task and resource environment cheaply, demanding only a few seconds per setup. A visualisation of the time/accuracy trade-off for different mehods is shown in Appendix D.

Table 2: Mean ± std. accuracies on NAS-Bench-201 datasets. Baselines are taken directly from Dong & Yang (2020), averaged over 500 runs (3 for weight-sharing methods). Search times are given for a CIFAR-10 search on a single 1080Ti GPU. NASWOT CIFAR10-search refers to searching on the CIFAR-10 dataset and then evaluating the final model on an alternative dataset. Performance of our training-free approach is given for different sample size N (also 500 runs). We also report the results for picking a network at random, and the best possible network from the sample. † This is our own implementation of REA for a fair comparison against AREAL (50 runs each).

| Method | Search (s) | CIFAR-10 | | CIFAR-100 | | ImageNet-16-120 | |
|---|---|---|---|---|---|---|---|
| | | validation | test | validation | test | validation | test |
| **Non-weight sharing** | | | | | | | |
| REA | 12000 | 91.19±0.31 | 93.92±0.30 | 71.81±1.12 | 71.84±0.99 | 45.15±0.89 | 45.54±1.03 |
| RS | 12000 | 90.93±0.36 | 93.70±0.36 | 70.93±1.09 | 71.04±1.07 | 44.45±1.10 | 44.57±1.25 |
| REINFORCE | 12000 | 91.09±0.37 | 93.85±0.37 | 71.61±1.12 | 71.71±1.09 | 45.05±1.02 | 45.24±1.18 |
| BOHB | 12000 | 90.82±0.53 | 93.61±0.52 | 70.74±1.29 | 70.85±1.28 | 44.26±1.36 | 44.42±1.49 |
| **Weight sharing** | | | | | | | |
| RSPS | 7587 | 84.16±1.69 | 87.66±1.69 | 59.00±4.60 | 58.33±4.34 | 31.56±3.28 | 31.14±3.88 |
| DARTS-V1 | 10890 | 39.77±0.00 | 54.30±0.00 | 15.03±0.00 | 15.61±0.00 | 16.43±0.00 | 16.32±0.00 |
| DARTS-V2 | 29902 | 39.77±0.00 | 54.30±0.00 | 15.03±0.00 | 15.61±0.00 | 16.43±0.00 | 16.32±0.00 |
| GDAS | 28926 | 90.00±0.21 | 93.51±0.13 | 71.14±0.27 | 70.61±0.26 | 41.70±1.26 | 41.84±0.90 |
| SETN | 31010 | 82.25±5.17 | 86.19±4.63 | 56.86±7.59 | 56.87±7.77 | 32.54±3.63 | 31.90±4.07 |
| ENAS | 13315 | 39.77±0.00 | 54.30±0.00 | 15.03±0.00 | 15.61±0.00 | 16.43±0.00 | 16.32±0.00 |
| **Training-free** | | | | | | | |
| NASWOT (N=10) | 3.8 | 89.13 ± 1.26 | 92.28 ± 1.13 | 68.19 ± 1.97 | 68.32 ± 1.99 | 39.11 ± 4.39 | 38.80 ± 4.73 |
| NASWOT (N=100) | 28.73 | 90.54 ± 0.83 | 93.36 ± 0.72 | 70.15 ± 1.37 | 70.31 ± 1.38 | 39.60 ± 2.98 | 39.21 ± 3.23 |
| NASWOT CIFAR10-search (N=100) | | | | | | 43.67 ± 1.73 | 43.74 ± 1.81 |
| NASWOT (N=1000) | 429.2 | 90.97 ± 0.3 | 93.75 ± 0.32 | 70.85 ± 1.02 | 71.09 ± 0.96 | | |
| Random | N/A | 83.20 ± 13.28 | 86.61 ± 13.46 | 60.70 ± 12.55 | 60.83 ± 12.58 | 33.34 ± 9.39 | 33.13 ± 9.66 |
| Optimal (N=10) | N/A | 89.92 ± 0.75 | 93.06 ± 0.59 | 69.61 ± 1.21 | 69.76 ± 1.25 | 43.11 ± 1.85 | 43.30 ± 1.87 |
| Optimal (N=100) | N/A | 91.05 ± 0.28 | 93.84 ± 0.23 | 71.45 ± 0.79 | 71.56 ± 0.78 | 45.37 ± 0.61 | 45.67 ± 0.64 |
| REAL † | 12000 | - | 93.59 ± 0.53 | - | 70.84 ± 1.41 | - | 44.90 ± 1.16 |
| AREAL | 12000 | - | 93.80 ± 0.29 | - | 71.30 ± 1.27 | - | 44.84 ± 1.05 |

## 5 Assisted Regularised EA - AREA

Our proposed score can also be incorporated into existing NAS algorithms. To demonstrate this we implemented a variant of REA (Real et al., 2019), which we call Assisted-REA (AREA). REA starts with a randomly-selected population (in our experiments the population size is 10). AREA instead randomly-samples a larger population (in our experiments 1000) and uses our score to select the initial population for the REA algorithm. Pseudocode can be found in Appendix E. When implementing AREA using the NAS-Bench-201 API the precise epoch accuracy used was not specified in Dong & Yang (2020) for their REA results across all datasets. Therefore we opted to use 'ori-test@12' for this benchmark. We denote the algorithms REAL and AREAL (for *Local* implementations) to distinguish them from those using the settings in Dong & Yang (2020). Tables 1 and 2 shows AREAL improves upon REAL in most cases across NAS-Bench-101 and NAS-Bench-201.

## 6 Conclusion

NAS has previously suffered from intractable search spaces and heavy search costs. Recent advances in producing tractable search spaces, through NAS benchmarks, have allowed us to investigate if such search costs can be avoided. In this work, we have shown that it is possible to run a search algorithm—NASWOT—in a matter of seconds, relying on simple, intuitive observations made on initialised neural networks, that challenges more expensive black box methods. We also demonstrate how our score can be combined into an existing NAS algorithm. We find our score-assisted algorithm—AREA—performs better than the original version on a number of benchmarks. This work is not without its limitations; our scope is restricted to convolutional architectures for image classification on benchmark search spaces. However, we hope that this will be a powerful first step towards removing training from NAS and making architecture search cheaper, and more readily available to practitioners.

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

# A HISTOGRAMS OF CORRELATIONS

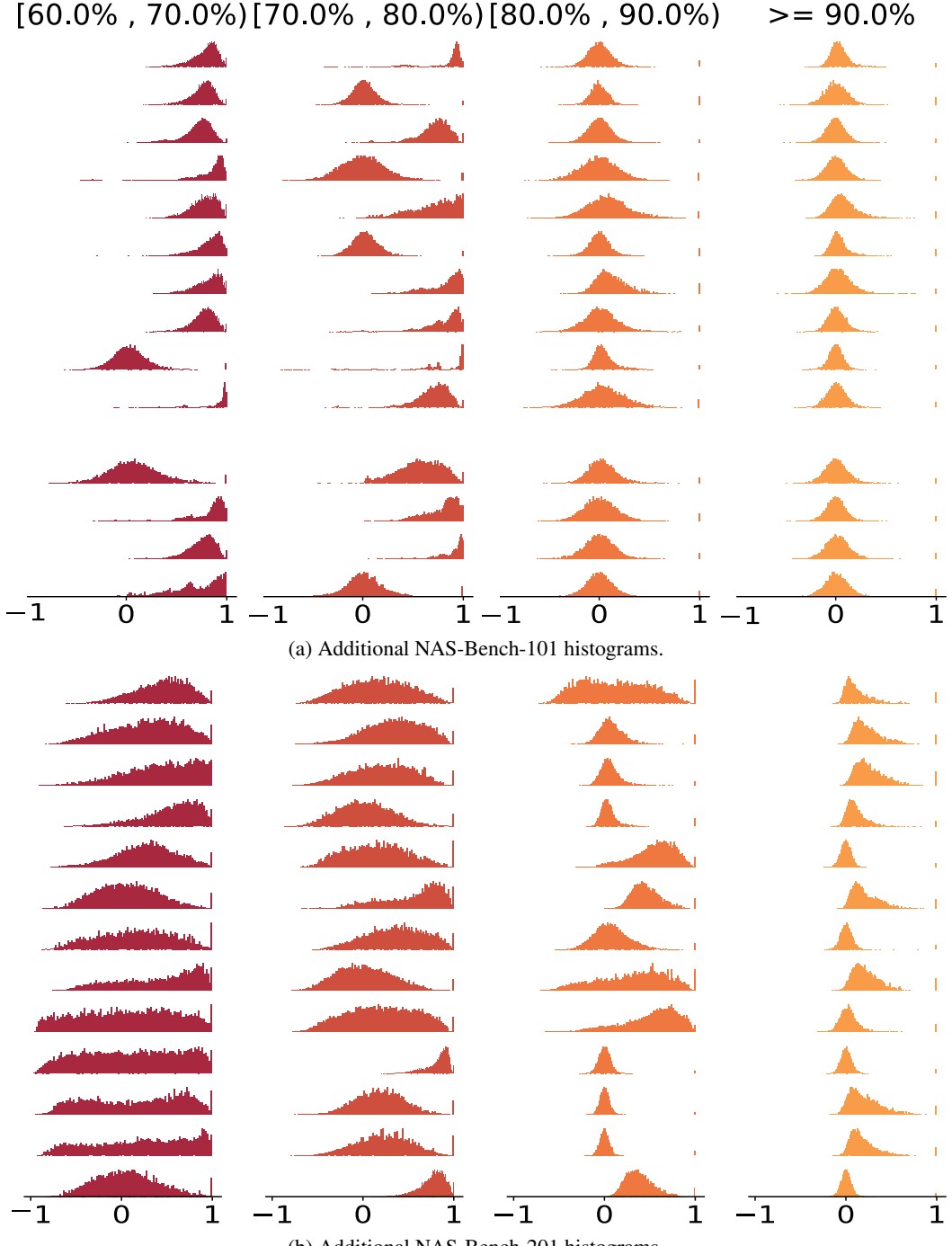

(a) Additional NAS-Bench-101 histograms.

(b) Additional NAS-Bench-201 histograms.

# B  NAS BENCHMARKS

## B.1  NAS-BENCH-101

In NAS-Bench-101, the search space is restricted as follows: algorithms must search for an individual cell which will be repeatedly stacked into a pre-defined skeleton, shown in Figure 6c. Each cell can be represented as a directed acyclic graph (DAG) with up to 9 nodes and up to 7 edges. Each node represents an operation, and each edge represents a state. Operations can be chosen from: $3 \times 3$ convolution, $1 \times 1$ convolution, $3 \times 3$ max pool. An example of this is shown in Figure 6a. After de-duplication, this search space contains 423,624 possible neural networks. These have been trained exhaustively, with three different initialisations, on the CIFAR-10 dataset for 108 epochs.

## B.2  NAS-BENCH-201

In NAS-Bench-201, networks also share a common skeleton (Figure 6c) that consists of stacks of its unique *cell* interleaved with fixed residual downsampling blocks. Each cell (Figure 6b) can be represented as a densely-connected DAG of 4 ordered nodes (A, B, C, D) where node A is the input and node D is the output. In this graph, there is an edge connecting each node to all subsequent nodes (A→ B, A→ C, A→ D, B→ C, B→ D, C→ D) for a total of 6 edges, and each edge can perform one of 5 possible operations (Zeroise, Identity, $3 \times 3$ convolution, $1 \times 1$ convolution, $3 \times 3$ average pool). The search space consists of every possible cell. As there are 6 edges, on which there may be one of 5 operations, this means that there are $5^6 = 15625$ possible cells. This makes for a total of 15625 networks as each network uses just one of these cells repeatedly. The authors have manually split CIFAR-10, CIFAR-100, and ImageNet-16-120 (Chrabaszcz et al., 2017) into train/val/test, and provide full training results across all networks for (i) training on train, evaluation on val, and (ii) training on train/val, evaluation on test. The split sizes are 25k/25k/10k for CIFAR-10, 50k/5k/5k for CIFAR-100, and 151.7k/3k/3k for ImageNet-16-120. We obtained these datasets via the NAS-Bench-201 repository (Dong, 2020).

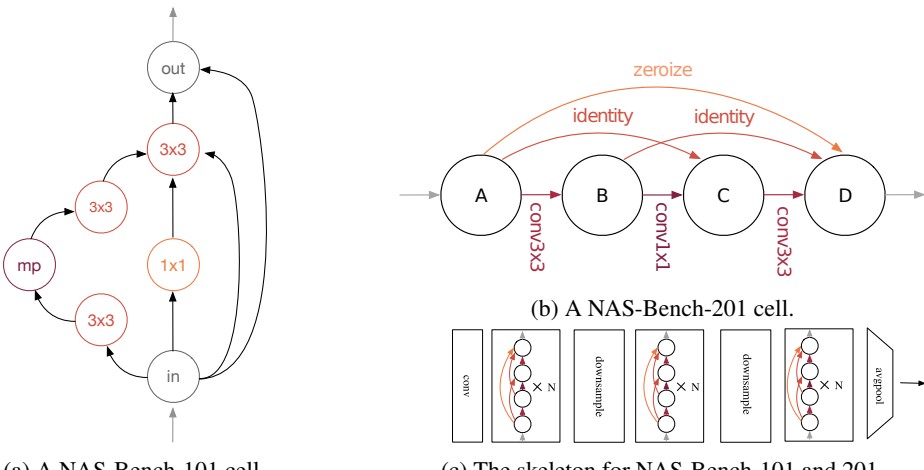

(a) A NAS-Bench-101 cell.

(b) A NAS-Bench-201 cell.

(c) The skeleton for NAS-Bench-101 and 201.

Figure 6: (a): An example cell from NAS-Bench-101, represented as a directed acyclic graph. The cell has an input node, an output node, and 5 intermediate nodes, each representing an operation and connected by edges. Cells can have at most 9 nodes and at most 7 edges. NAS-Bench-101 contains 426k possible cells. By contrast, (b) shows a NAS-Bench-201 (Dong & Yang, 2020) cell, which uses nodes as intermediate states and edges as operations. The cell consists of an input node (A), two intermediate nodes (B, C) and an output node (D). An edge e.g. A→ B performs an operation on the state at A and adds it to the state at B. Note that there are 6 edges, and 5 possible operations allowed for each of these. This gives a total of $5^6$ or 15625 possible cells. (c): Each cell is the constituent building block in an otherwise-fixed network skeleton (where N=5). As such, NAS-Bench-201 contains 15625 architectures.

## C Ablation Figures

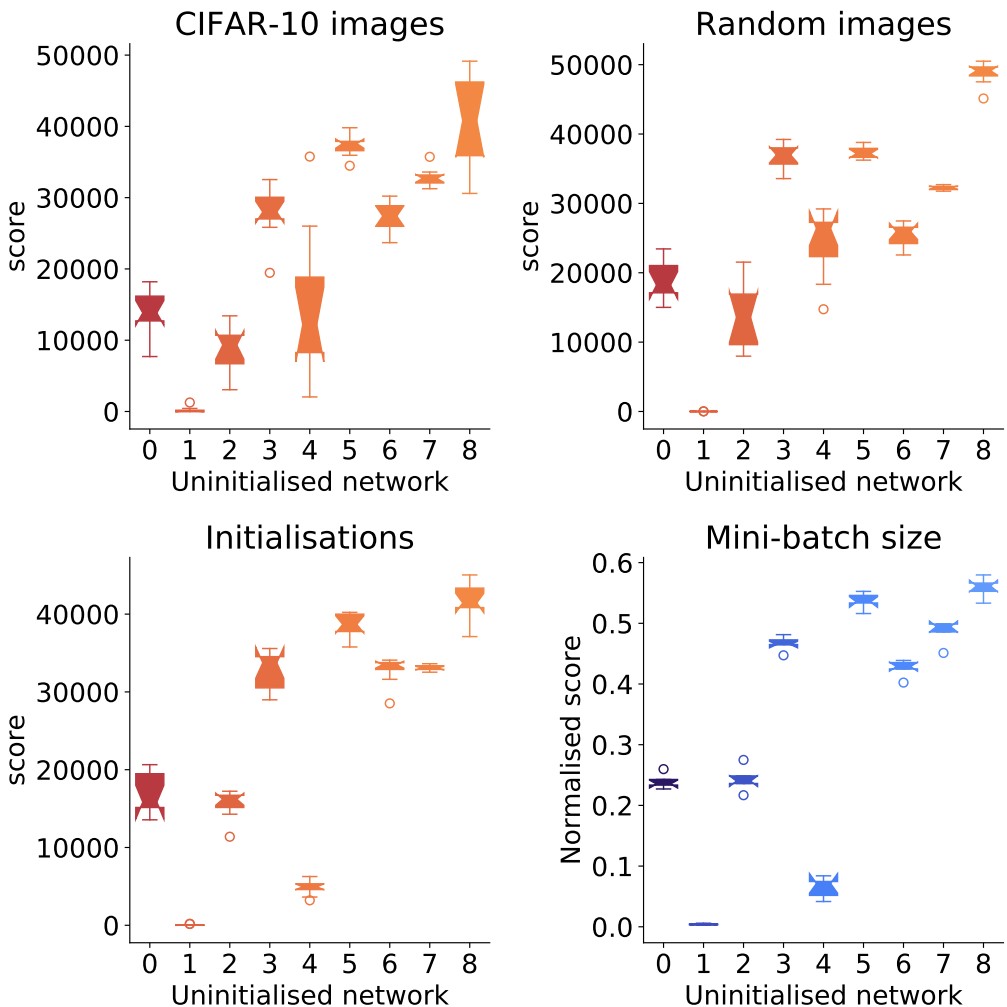

Figure 7: Effect of different CIFAR-10 image (top-left), initialisation (bottom-left), random input images (top-right), and mini-batch sizes (bottom-right) for 9 randomly selected NAS-Bench-201 architectures (one in each 10% percentile range from 10-20, ..., 90-100). Random input data are normally distributed. For each network 10 samples were taken for each ablation, using a mini-batch size of 256, except for mini-batch size for which mini-batch sizes of 32, 64, 128, 256, and 512 were used. There is less variance in score when using different random input images as opposed to different CIFAR-10 images.

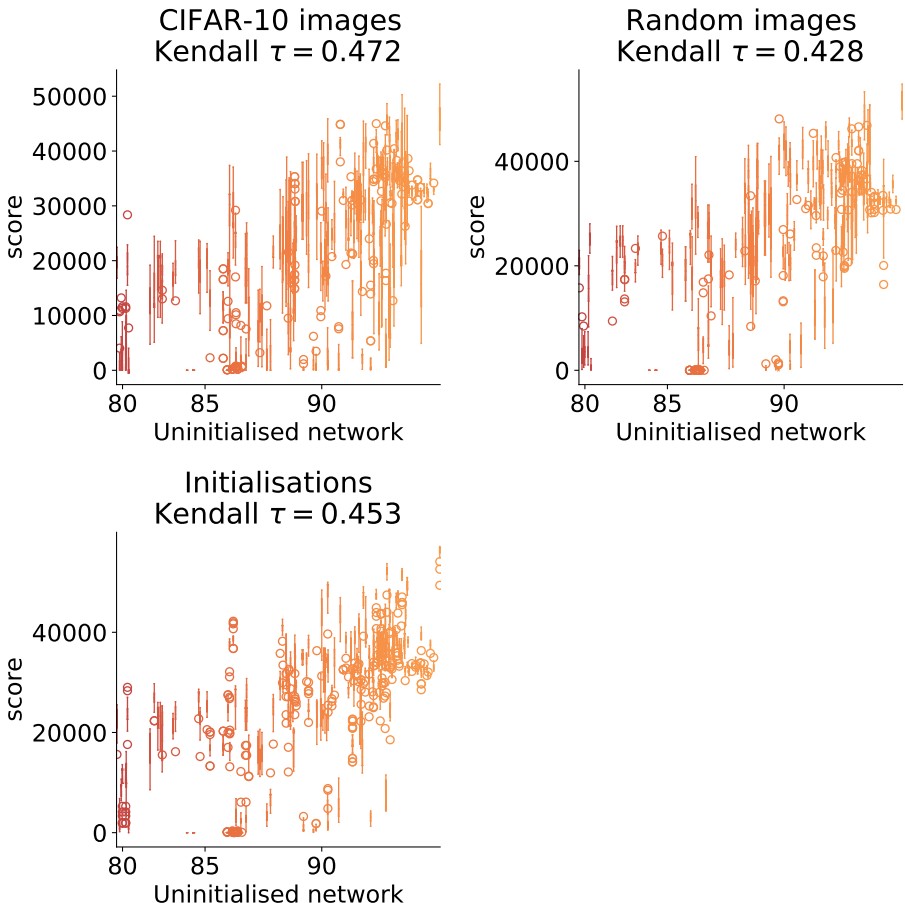

Figure 8: Further ablation showing the effect of different CIFAR-10 image (top-left), initialisation (bottom-left), random input images (top-right), and mini-batch sizes (bottom-right) for 100 randomly selected NAS-Bench-201 architectures in the 80% + CIFAR-10 accuracy range when using cutout as the augmentation method when calculating the score. For each network 20 samples were taken for each ablation.

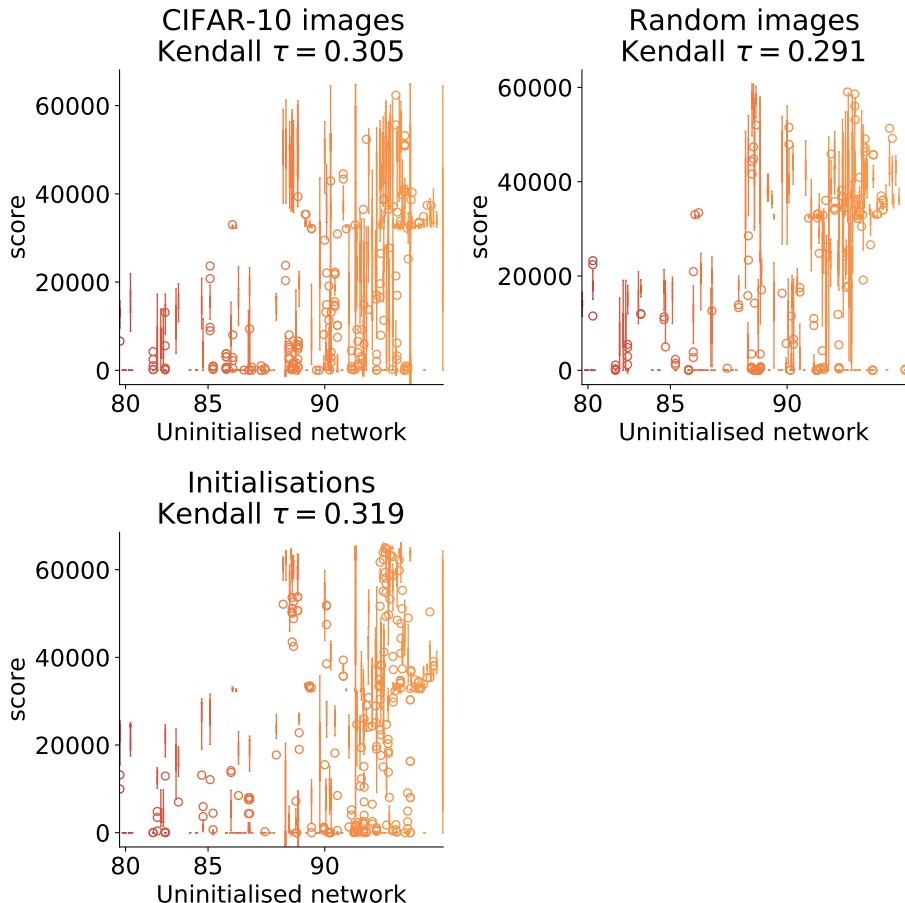

Figure 9: Further ablation showing the effect of different CIFAR-10 image (top-left), initialisation (bottom-left), random input images (top-right), and mini-batch sizes (bottom-right) for 100 randomly selected NAS-Bench-201 architectures in the 80% + CIFAR-10 accuracy range when using additive Gaussian noise as the augmentation method when calculating the score. For each network 20 samples were taken for each ablation.

# D  SEARCH TIME COMPARISON

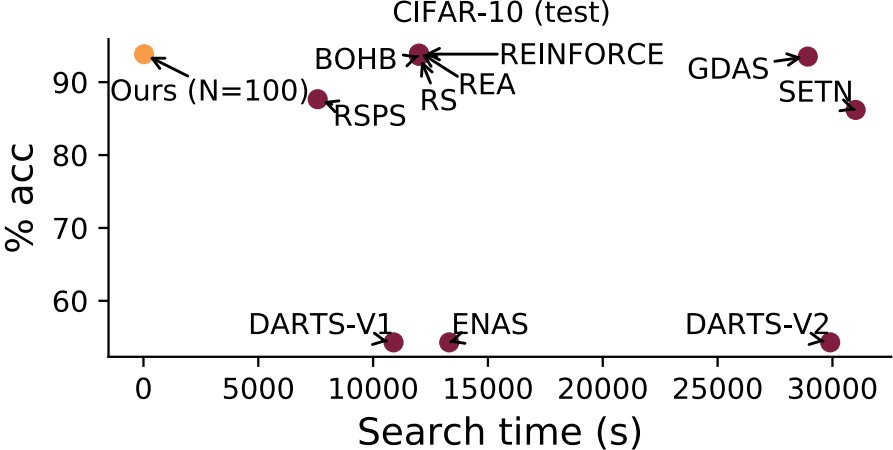

Figure 10: Visualising the tradeoff between search time and accuracy on CIFAR-10 (test) for different NAS algorithms on NAS-Bench-201. By removing the need for training, our method is able to find accurate networks in seconds instead of hours.

# E  AREA ALGORITHM

---
**Algorithm 3** Assisted Regularised EA - AREA
---

population = []
generator = RandomGenerator()
**for** i=1:M **do**
    net = generator.generate()
    scored_net = net.score()
    population.append(scored_net)
Keep the top N scored networks in the population
history = []
**for** net in population **do**
    trained_net = net.train()
    history.append(trained_net)
**while** time limit not exceeded **do**
    Sample sub-population, S, without replacement from population
    Select network in S with highest accuracy as parent
    Mutate parent network to produce child
    Train child network
    Remove oldest network from population
    population.append(child network)
    history.append(child network)
chosen_net = Network in history with highest accuracy

---

# F VARYING INITIALISATIONS

In Figure 11 we examine the effect of different initialisation functions on the efficacy of our score. We find that the scoring function is robust to many different intialisation strategies, though it fails for Uniform(0,1) initialisation. In Figure 12 we confirm that this effect is similar when the score is aggregated over multiple re-initialisations of the network.

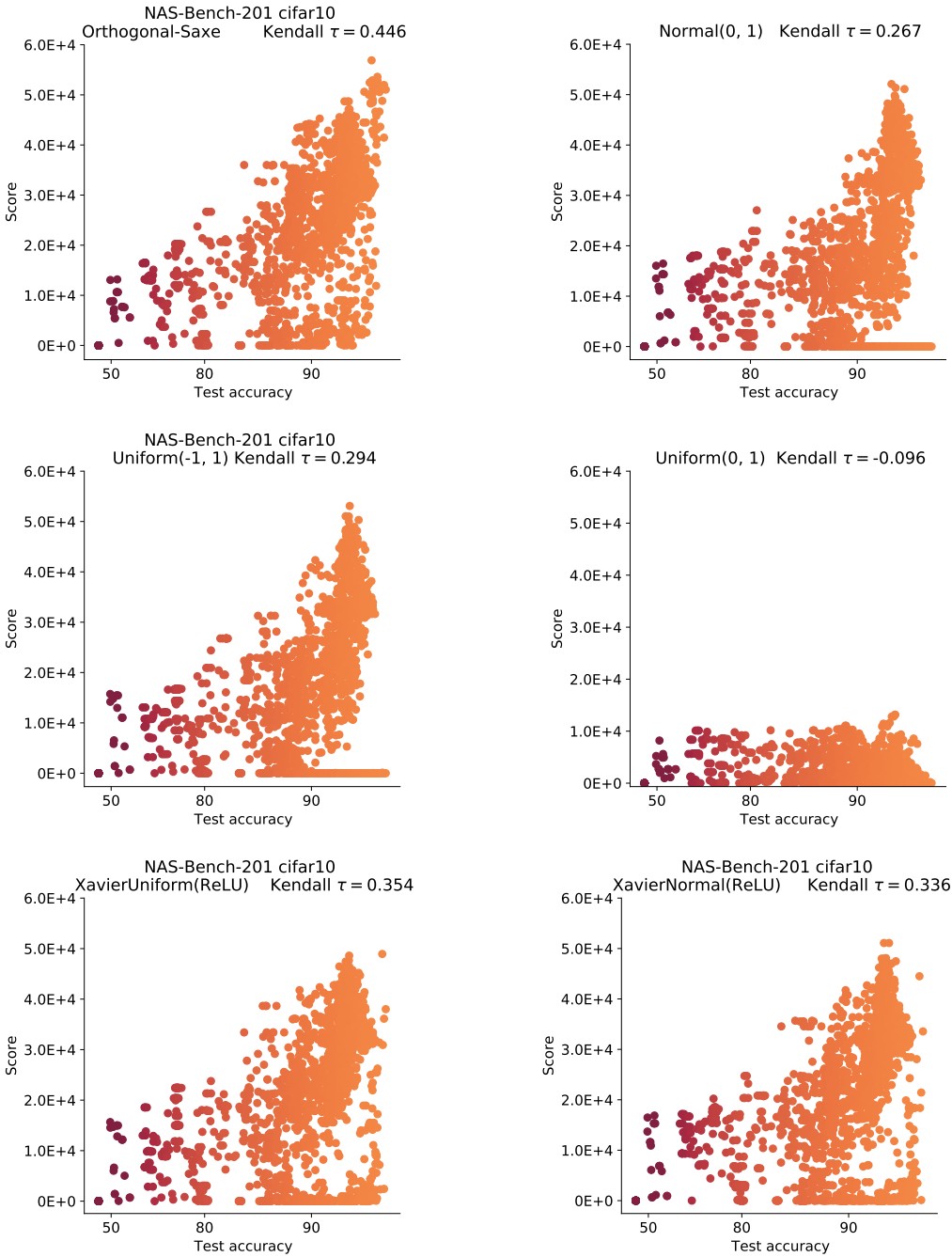

Figure 11: Plots of association between score and final CIFAR-10 validation accuracy for various initialisation strategies.

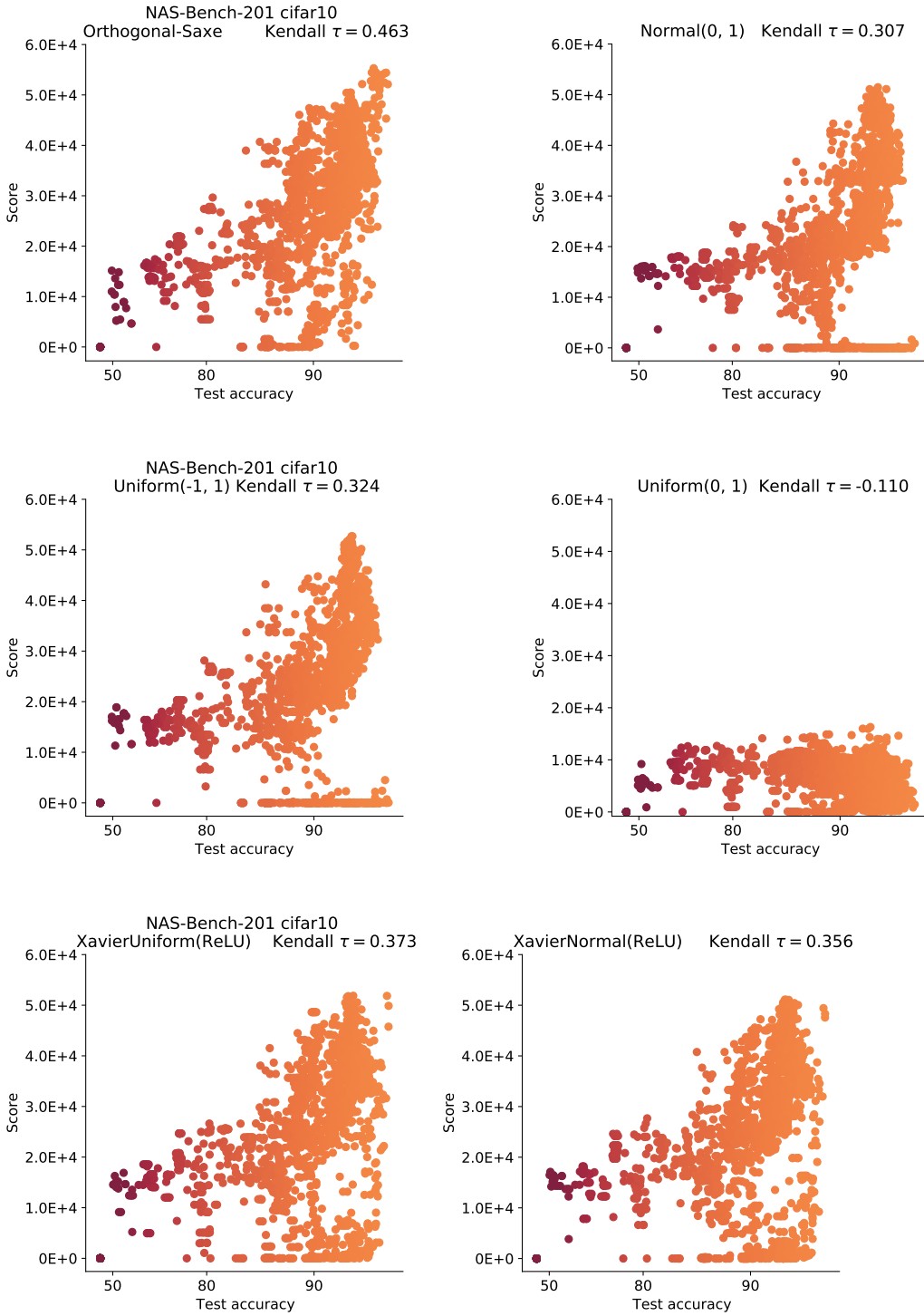

Figure 12: Plots of association between aggregated score and final CIFAR-10 validation accuracy for various initialisation strategies. The aggregated score is the maximum score of 10 independent initialisations of the same architecture.

# G VARYING $\beta$

The scoring mechanism introduced in Section 3 relies on a cutoff value $\beta$ that decides which portion of the histogram of correlations we look at. Given that our goal is to assign high scores to histograms with a small positive skew, we set the default value of $\beta = 0.25$. In Figure 13 we investigate different values of $\beta$, reporting the CIFAR-10 test accuracy for networks in NAS-Bench-101 and NAS-Bench-201.

We linearly sample five values of $\beta$ between 0.025 and 1 and report mean accuracy over 500 runs. We also show the standard deviation in accuracy for each $\beta$ in the shaded area. The figure shows that in both benchmarks, the scoring mechanism is reasonably robust to different choices of $\beta$, but deteriorates at very large values.

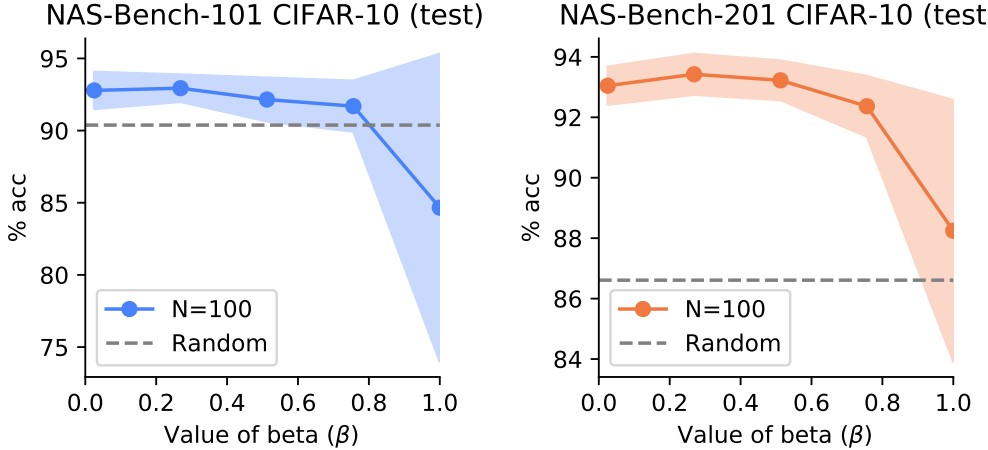

Figure 13: Controlling for different values of $\beta$ in the scoring function. Points are shown as the mean value over 500 runs, with the standard deviation shown in the shaded areas.

# H  SCORE BY EPOCH

We evaluated the score during training and plotted the trajectory of the score for 100 networks for NAS-Bench-201 trained using the CIFAR-10 dataset in Figure 14.

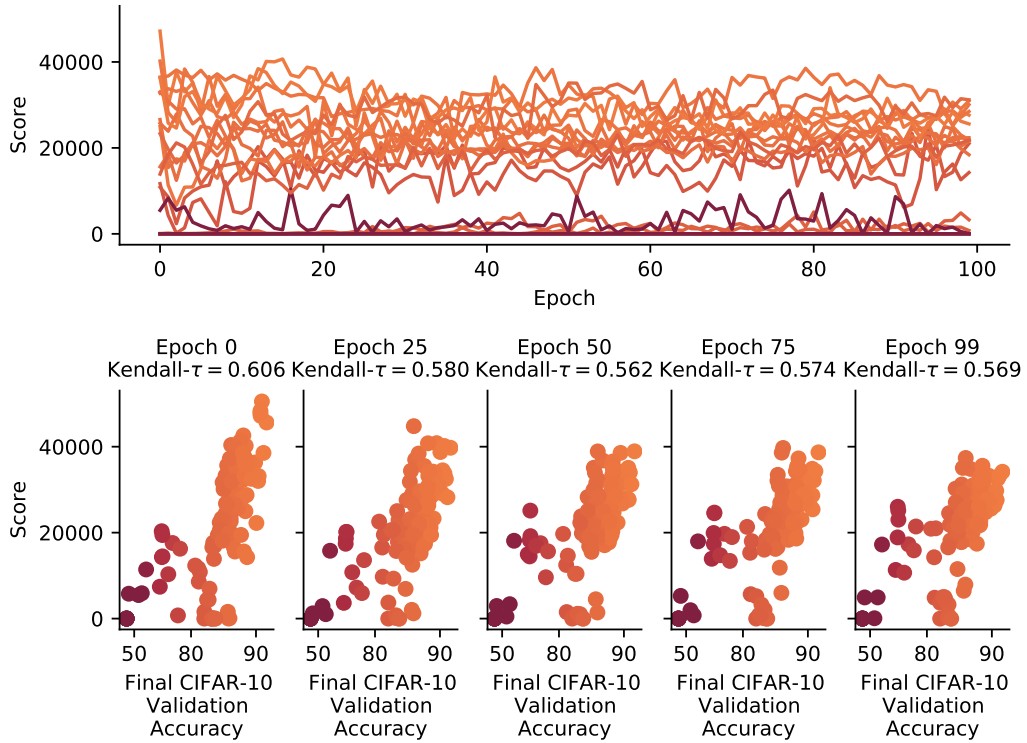

Figure 14: (top) A plot of score trajectories during training for 20 networks in NAS-Bench-201. (bottom) Plots of final CIFAR-10 Validation accuracy compared to score for 125 networks for epoch 0, 25, 50, 75, and 99. The correlation between score and final test accuracy reduces marginally with training epoch but an association between the two quantities remains.

# I  AUGMENTING WITH GAUSSIAN NOISE AND COLOUR JITTER

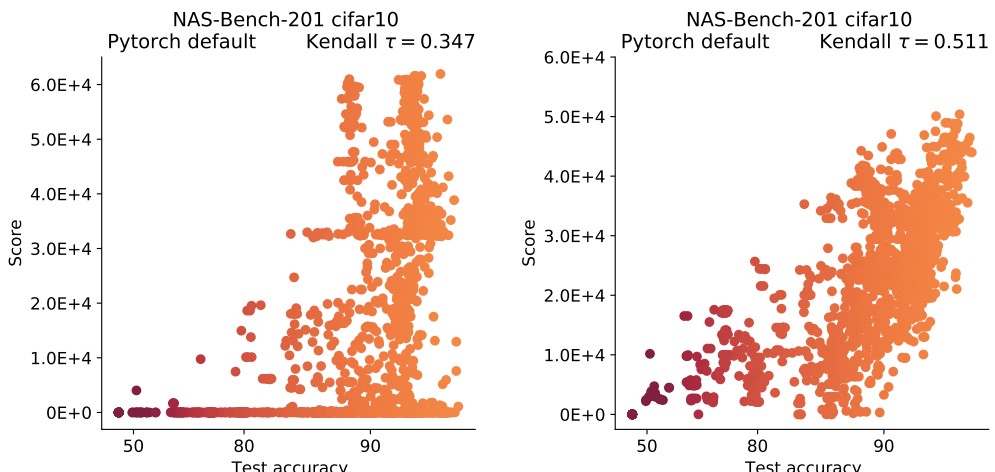

Figure 15: Plots of CIFAR-10 final accuracy against score for alternative augmentation types: (left) Additive gaussian noise $\mathbf{N}(0, 0.01)$, (right) colour jitter.

# J  FURTHER RANDOM IMAGE EXPERIMENTS

We ran further evaluations of the score for random images for 6 seeds. These are shown in Figure 16

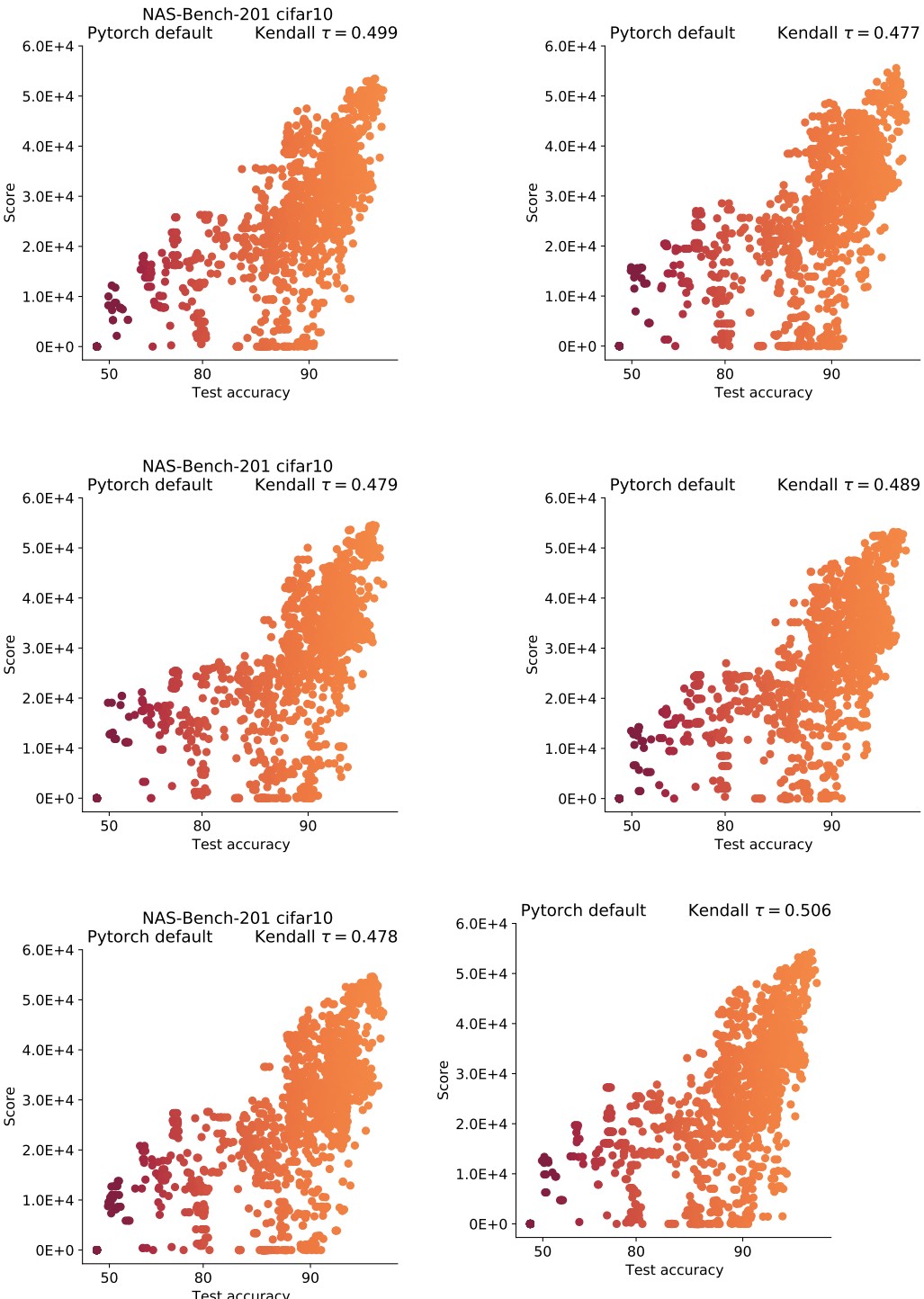

Figure 16: Plots of association between score and final CIFAR-10 validation accuracy for 6 seeds using random images (generated via torchvision FakeData).

