# OpenReview forum: "Neural Architecture Search without Training"
_ICLR.cc/2021/Conference — Reject_

### Official Review · AnonReviewer1 · 2020-10-28
**Interesting albeit under-explored proof-of-concept for training-free NAS.**

**Rating:** 6
**Confidence:** 4

**Review:**

Summary:
The authors propose a training-free way of estimating the performance of a deep net architecture after training using correlations between linearizations of the network at initialization for different augmentations of the same image. This estimate is used as signal to construct NAS algorithms that do not require training deep nets and is evaluated on two datasets. Although I have significant concerns about the practicality of the method, I believe it establishes a sufficiently distinct direction for NAS research that could merit acceptance.

Strengths:
1. To my knowledge the paper is the first to implement training-free NAS.
2. The method is simple and easy to implement.
3. The method achieves decent performance on CIFAR in dramatically less time than previous approaches.
4. What seems like a complete codebase is provided (although see Question 1 below).
5. The paper is clear and easy-to-follow.

Weaknesses:
1. The justification for the actual score used is weak (see Questions 2-3 below).
2. The method seems limited to vision data and networks with ReLU activations.
3. Performance on the ImageNet subset of NAS-Bench-201, the only evaluation using non-CIFAR data, is poor.
4. Limited exploratory and benchmark evaluations (see Questions 4-7 below).

Questions:
1. I was not able to run the provided code (search.py) using the instructions provided; could the authors provide a dependency list?
2. As the score used is non-obvious and has no mathematical basis, it seems likely some trial-and-error was used to find it; is this the case, and if so what sorts of rules were tried that did not work.
3. Why not just use the correlations between gradients rather than using the indicator function to obtain some linearization?
4. What is the effect of beta on performance?
5. Do other types of data augmentation work?
6. For NAS-Bench-201, why was N>100 (e.g. N=1000) not tried? There is clearly room to improve and going that high still leaves the NASWOT algorithm by far the fastest.
7. How does NASWOT perform on larger search spaces such as DARTS (Liu et al., 2019)?

Notes:
1. In two locations in the paper (3rd para of intro, 3rd para of background) the authors suggest that Li & Talwalkar (2019) show that WS inhibits architecture search and/or struggles against random search; however, that paper also shows that combining WS with random search is outperforms the latter.
2. “Moreover, popular search spaces have been shown to be over-engineered, exhibiting little variety in their trained networks (Yang et al., 2020).” - is there evidence that NAS-Bench-101 and NAS-Bench-201 do not also suffer from this? Both were released before the publication of Yang et al. (2020).
3. “Given a neural network with rectified linear units, we can, at each unit in each layer, identify a binary indicator as to whether the unit is inactive (the value is negative and hence is multiplied by zero) or active (in which case its value is multiplied by one).” - is the proposed method dependent on ReLU activations being used?
4. Figure 4: what is the small circle that appears either above or below many of the box-and-whisker points?
5. Table 2: it is standard to report the optimal in the entire search space, not just for N=10/100.

# Post-response update
Thank you to the authors for very helpful clarifications. This paper provides a reasonable start for a new potential direction in NAS research and so may be worth presenting at the conference, but the justification and applicability of the method is somewhat limited. I therefore stand by my original assessment.

---

> ### Author Response · Authors · 2020-11-19
> **Author Response**
>
> We would first like to thank the reviewer for their helpful comments, feedback, and detailed questions. We will try to address each of their concerns here:
>
> Dependency list: we apologise for not including these and have added it to the source code zip file as env.yml. Please let us know if you have any issues.
>
> Trial-and-error: The score has 2 parts: first is the calculation of the intra-batch correlation between input Jacobians and the second is summarising the information in the correlation matrix into a scalar value. Visual inspection of the correlation matrices as shown in Figure 1 steered the choice of minibatch and augmentation. The second part of the score was also derived from the visual exploration shown in Figure 1 in the paper (i.e. we tried to design a score that could capture the difference in histograms between poor and strong networks).  We wanted a score without any learnable parameters. We have included ablation of the main parameter beta in Appendix G.
>
> Why indicator rather than directly using correlations between gradients: In Figure 3 we can see that the best networks have correlation histograms with low variance and a small positive skew. The indicator function is important to capture this small positive skew, without it we may pick up on networks that have low variance but are centered around or below zero.
>
> Effect of beta on performance: Overall the effect of changes to beta is quite minimal unless beta is set to be very close to 1. We have added an analysis of this in Appendix G of the paper.
>
> Other types of data augmentation: The specific choice of data augmentation shouldn’t impact the efficacy of our method - the most important consideration is that we are measuring the models responses to small perturbations of one image. In Appendix I we have added two new augmentation types, additive Gaussian noise and colour jitter, to illustrate this.
>
> Why no N>100?: In an unbiased sample of 100 networks, we’re almost guaranteed to have networks across the full spectrum of accuracies. Therefore, adding more to the sample has diminishing returns. We have, however, added N=1000 to the table in the paper, which gives a very slight improvement in accuracy across datasets.
>
> Performance on the ImageNet subset of NAS-Bench-201: For clarification, ImageNet16-120 is not the same as ImageNet and contains images of size 16x16. Our drop in performance can be explained by a misunderstanding of the NAS-Bench-201 procedure. Our original results show the performance by performing the search using the same dataset as is used for the evaluation of the found network. The protocol used by NAS-Bench-201 was actually to search using CIFAR-10 and only evaluate the final model using ImageNet16-120. When we do this the average performance is much more competitive.
>
> Small circles in box and whisker plots: The small circles are outliers. For lower quartile L, upper quartile U, and interquartile range I, outliers are defined as points < L - 1.5*I and > U + 1.5*I.

---

### Official Review · AnonReviewer2 · 2020-10-28
**Official Blind Review #2**

**Rating:** 4
**Confidence:** 4

**Review:**

### Summary
This paper proposes a zero-shot NAS algorithm.  Instead of using validation accuracy that requires training, a score **S** is defined. It is computed as follows:
    A single image is augmented using cut-out several times to produce a batch of images. For every datapoint, the Jacobian is computed to form a vector J. Then the correlation matrix $\Sigma$ for the J is computed. Score S is the total number of entries in $\Sigma$ between 0 and upper bound $\beta$. $\beta$ is a hyper parameter.
The search algorithm **NASWOT** now randomly samples K networks from the search space and finds the network with the highest score **S**.

###Pros:
  The idea of using a metric based on Jacobian rather than validation accuracy is interesting.
   Their experimental setup demonstrated the correlation between ranking based on S and validation accuracy, NAS on 3 datasets on NASBENCH-201

###Questions

 1. Main concern is that as the method relies on only 1 image and the initialization, what is the guarantee that it would generalize.  In Figure 4, the ablation study considers networks from different accuracy buckets. The Standard deviation of the score is high for some networks.  What would be more useful is if around 200 networks were sampled from 80 to 100 accuracy bucket, for each of the 20 images (instead of 10), we compute the Kendall Tau of the ranking based on **S** and validation accuracy. You can report the average Kendall Tau. Similar experiment can be performed for initialization too.

2. How will this method account for the fact that some datasets have very diverse images? For example, NASWOT deteriorates in performance on Imagenet which has more diverse images. Kendall Tau computed on ImageNet in Figure 3 also reflects the same. Instead of 1 image, can we sample 10 representative images from the distributed and compute average of S on all of them?

3.  In their paper, Real et al. use an initial population of 100. 10 might be too less for initial population. So the comparison of REAL vs AREAL might not be fair. Even in that case, the performance improvement is not significant.

4. In figure 3, why is the correlation not higher than 0.55? Does this mean that while it has some signal, it is not enough to discriminate well performing networks from others?

---

> ### Author Response · Authors · 2020-11-19
> **Author Response**
>
> We thank the reviewer for their comments and suggestions. We are glad they found our work interesting. We will now address their questions.
>
> Generalisation/reliance on a single image: the reviewer suggested that Figure 4 could be strengthened by using networks from the 80-100% accuracy range and taking the Kendall Tau of the ranking and validation accuracy over 20 images (instead of 10). In light of this we have added further ablations using generated images where pixel values are randomly sampled (using torchvision FakeData) in Appendix J (Figure 16) We believe the method is not specific to the dataset per se but that the score captures a property of the network structure.
>
> Scaling to datasets with more diverse images: It’s worth highlighting that the “ImageNet” used in our experiments is a downsampled version of “ImageNet-16” as introduced in NAS-Bench-201. A subset of 200 classes is taken, and the images are downsampled to 16$\times$16 pixels.
>
> REA uses population size 100: Our implementation of REA was taken directly from NAS-Bench-201.
>
> Figure 3 correlation is only 0.55: It’s true that there is some noise in the scoring mechanism. Given that there are several thousands of networks in each NAS benchmark and that our approach incorporates several random steps (sampling an image, adding random noise etc.), it’s unlikely that a simple scoring function could ever give a noise-free ranking. However, we demonstrated in our experimental section that the correlation of 0.55 was enough of a signal to perform rapid and effective NAS across both NAS-Bench-101 and NAS-Bench-201.

---

### Official Review · AnonReviewer3 · 2020-10-29
**Need more analysis of the proposed method**

**Rating:** 5
**Confidence:** 4

**Review:**

# Summary
This paper attempts to infer a network's accuracy at initialization without training it, which can speed up neural architecture search and greatly reduce the search cost. Specifically, they propose a metric based on the Jacobian of the loss with respect to a minibatch of input data. The authors show that with this metric, they can find architectures with reasonable accuracy on CIFAR-10/CIFAR-100 in the NAS-Bench-201, while using much less search cost compared to previous NAS methods.



# Strong points
1. This paper is exploring a novel and interesting direction. Estimating the network’s performance at initialization can greatly reduce the search speed.
2. It might be difficult for the training-free metric to outperform conventional metrics (e.g., validation accuracy after training). This paper finds a good use case of the training-free metric in practice. The authors propose to use the training-free metric to select the initial population of evolutionary algorithms and empirically demonstrate its usefulness on CIFAR-10 and CIFAR-100.


# Weak points
1. It will be helpful to make the motivation of the proposed metric more clear. I found the second paragraph in Section 3 hard to understand without digging into the cited literature.

2. How to interpret the value in the correlation matrix \Sigma? Empirically, Figure 1 shows that we want to find an architecture whose values in matrix \Sigma are mostly around zero with a small positive skew. But how does this relate to the two motivations mentioned in Section 3: (1) being flexible and (2) being invariant/robust to small perturbations. Does small value mean less or more flexible/robust? This is important for people to understand why the metric works.

3. In the experiments, all the images in the minibatch are data augmentations of the same image. This seems to be an important design choice. More insights on this part would be very helpful. How is this design choice (using data augmentations of the *SAME* image) related to the two motivations (flexible & invariant)?

4. As mentioned by the authors, the two motivations are actually antagonistic. How does the proposed method balance them?

5. Why choose cutout as the data augmentation strategy? Is the method sensitive or not to other perturbations, e.g., adding small noise?

6. I notice that the correlation (tau) value in Figure 3 is not high. What is the correlation between the typical criteria (e.g., validation accuracy after training a small number of epochs) and the final test accuracy?

  The tau in Figure 3 is actually undefined. Is that Kendall tau?

  Why evaluate the correlation between **validation** accuracy, not the final **test** accuracy, as provided in NAS-Bench-201?

7. Section 5 mentions that “ori-test@12” is used as the metric during search. Is that the accuracy on the **test** set after 12 epoch training (with learning rate decay to 0)? But the common practice is to use the **validation** accuracy during search and report the final test accuracy. This seems to be unfair.

8. For NASWOT (N=100), the accuracy on CIFAR-10 is reasonable and actually pretty good considering the small search cost. However, the accuracy in ImageNet-16-120 seems to be very low. Why does this happen? Is it possible that the proposed method overfit to CIFAR-10?

  If I understand correctly, the architecture is searched on CIFAR-10, and then evaluated on all three datasets. What if we search on ImageNet-16-120 with this metric and can we match the performance of other baselines on ImageNet-16-120? This might be an unfair comparison, but will be important for people to know whether the proposed method/metric can generalize to different datasets.

9. Table 2 only reports N=10 and N=100. What if we significantly increase the value of N and will the performance be similar or better than other methods like RS and REINFORCE? As the proposed method uses very small cost, even using a large N, the cost would still be reasonable. It will be great to see we can achieve much better performance when increasing N.

10. The writing and organization in Section 4 need to be improved. Adding subtitles might make it easier to read. Also, Section 4 mentions REAL at the beginning but REAL is not explained in detail until Section 5. The rows “Optimal (N=10/100)” don’t seem to help validate the proposed method and are a bit confusing.

11. It will also greatly strengthen this work if the authors can show the effectiveness of the proposed metric on a more realistic search space, e.g., the DARTS search, and evaluate the found architecture on a larger dataset, e.g., ImageNet.

# Justification of rating
I like the idea of estimating a network’s performance without training it. But this paper needs more refinement before being accepted. As mentioned above, the motivation, and the relationship between the method and motivation, need to be explained more clearly. More analysis is needed to understand and justify the proposed method. The writing also needs improvement.

# After rebuttal
I would like to thank the authors for the hard work during the rebuttal. The ablation study of the data augmentation strategy and other added results are very helpful. Regarding the explanation of the flexibility and invariance, although I could get some intuition, I am still not fully convinced. So I keep my original rating. One possible way to make this work stronger and meet the acceptance criteria is to provide some empirical (or even better, theoretical) analysis of the influence of $\Sigma$ on the flexibility and invariance of a network.

---

> ### Author Response · Authors · 2020-11-19
> **Author Response**
>
> We thank the reviewer for their detailed comments and interest in the paper. The reviewer had many questions, which we will attempt to answer:
>
> Interpretation of values in $\Sigma$, Single image in minibatch, Balancing flexibility and invariance: These questions are highly related, so we will try to answer them in one go. As the reviewer mentions, there are two properties we are trying to capture with our score: (1) flexibility, and (2) invariance to perturbations. We can think of $\Sigma$ as describing the relationship between the linear maps produced by several small perturbations of one image (since every image in the minibatch is a perturbation of a single image). Small values of sigma (i.e. close to zero) imply that the network is very flexible - the correlations between maps are small (i.e. the network meets the first criterion). If the values of $\Sigma$ have little variance, this means that different perturbations of the image produced similar linear maps in the network (i.e. the network meets criterion (2), invariance to perturbations).
>
> To recap: values close to zero implies flexibility, small variance implies invariance to perturbations. For example, a network with correlations centred around 1 with small variance would have highly correlated linear maps (i.e. very inflexible, mapping all the images to the same linear operator), but invariance to perturbation. This would, quite rightly, score poorly, since considering all images to be the same gives us very few gradient directions to train in.
>
> We hope this clarifies the score. We understand that these two facets are quite subtle, and would be happy to answer any more questions the reviewer has!
>
> Why cutout?: Related to the above, cutout is used to test the networks ability to handle small perturbations to an image. In reality, the method is not specific to this choice of augmentation. To demonstrate this, we have added Appendix I. We  show the score when using additive gaussian noise and colour jitter. Both augmentations show some ability to rank networks, with colour jitter performing comparably to cutout.
>
> Correlation not high: The ordering that our score provides is by no means perfect, but we hope that the empirical results are strong enough to provide evidence that even a simple measure can be used to construct an interesting NAS technique. While not high, the correlation was enough to find good networks on both NAS benchmarks. We have added the definition of Tau to Figure 3, sorry for leaving this out.
>
> We correlated to validation accuracy so that we were using the validation set for every dataset (instead of test for CIFAR-10, and validation for the others). The correlation on the test set is almost identical as shown by Figure 4 of NAS-Bench-201.
>
> “ori-test@12”: This was chosen due to a misunderstanding of the NAS-Bench-201 on our part. Our algorithms search on the dataset for which they the found model is evaluated. NAS-Bench-201 does the search on CIFAR-10 irrespective of the evaluation dataset. We have now included a version of NASWOT that searches with CIFAR-10 and evaluates on ImageNet16-120 which actually improves the performance of the algorithm for that dataset. We used ori-test@12 with both the baseline - REAL - and our method  - AREAL - in order to be directly comparable.
>
> ImageNet-16-120 accuracy is low: we thank the reviewer for highlighting this. In fact, our original implementation did search directly on ImageNet-16-120. Since the images in CIFAR are larger, there is potentially more information in the CIFAR-minibatches of perturbations than the ImageNet ones. When we reran to match the implementation of NAS-Bench-201, our accuracy on ImageNet-16-120 improved to 43.67% validation accuracy. We have added this as a line in the table.
>
> Table 2 only reports N=10 and N=100: We have included results in the paper for N=1000. In general, we saw diminishing returns for adding more iterations. Assuming that we rank an unbiased sample of architectures, a sample size of 100 is likely to include architectures across the full spectrum of accuracies. Therefore when we add more comparisons, the ranking does not become more informative.
>
> Section 4 subtitles: We apologise for the lack of clear narrative in this section! We have added bold text so that the section now starts with the algorithm introduction, then gives NAS-Bench-101 results, then NAS-Bench-201 results. We hope this makes the section a little easier to read.

---

### Official Review · AnonReviewer4 · 2020-10-30
**Interesting, but important comparisons are missing**

**Rating:** 5
**Confidence:** 5

**Review:**

** Summary

The paper mainly introduces a metric to benchmark the performance of neural networks without training – the correlation of Jacobian subject to different augmented versions of a single image. The key motivation is, high-performance networks tend to represent data of small perturbations with different hyperplanes at initialization, so that the distinguishing capability may also be stronger. The divergence of the hyperplanes can be efficiently estimated via the correlation of the Jacobian, thus quantified by the score in Eq. 2. The effectiveness of the proposed metric is mainly verified in two NAS benchmarks (NAS-Bench-101 and NAS-Bench-201), whose correlation to the actual accuracy is relatively significant (Fig 3). Compared with existing NAS frameworks, the proposed method is very efficient, moreover, able to obtain competitive performance.

** Pros

I really appreciate the paper proposes a new direction to benchmark and understand neural networks. The motivation is very impressive. Previous NAS frameworks based on score predictors, e.g. NAO [*1] and ChamNet [*2], have shown that it is possible to directly predict the performance from the architecture embeddings without training, however, the underlaying mechanism is not clear. The proposed metric may help to uncover the relation between architecture choices and the performances. Furthermore, with the help of the metric, the proposed NAS framework (NASWOT) is elegant and seems to be effective, which could be useful in practice.

** Cons

1)	Lack of theoretical evidence to support the intuition of the proposed method. To my knowledge, network initialization is important but cannot interpret all the behaviors; so, I still doubt why and how the Jacobian at initialization reflects the final score. I guess [*3] may help in the analysis.
2)	Some important ablations are missing in the experiments, which makes the method less convincing. Please refer to the following suggestions.

** Concerns and suggestions

Major:

To my understanding, the methodology of initialization (such as identity init, random gaussian with fixed variance (e.g. 0.01), MSRA init. [*4], etc.) could affect the initial Jacobian a lot, but little on the final accuracy especially with BN, which is not compared in the paper (I suppose the comparison in Fig 7 (bottom-left) is performed with different runs rather than different initialization methods, please correct me if I make some misunderstandings). Notice that some initialization methods like [*4] may leak architecture parameters (e.g. number of input channels). I think it is very important to check whether the Jacobian under a certain initialization methodology (improperly) takes advantage of the architecture bias in NAS benchmarks. I am pleased to raise the rating if the authors clear my concern, for example, providing comparisons of different initialization methods, or benchmarking the method with different search spaces on large datasets (e.g. search space of FBNet [*5] on ImageNet).

Minor:

1)	It is interesting if the authors analyze or visualize (just like Fig 1) how the correlation of Jacobian evolves during the training for different architectures.
2)	Please clarify the details about “… adjust the final classifier layer to output a scalar” in Page 4.

[*1] Luo, Renqian, et al. "Neural architecture optimization." Advances in neural information processing systems. 2018.

[*2] Dai, Xiaoliang, et al. "Chamnet: Towards efficient network design through platform-aware model adaptation." Proceedings of the IEEE Conference on computer vision and pattern recognition. 2019.

[*3] Jacot, Arthur, Franck Gabriel, and Clément Hongler. "Neural tangent kernel: Convergence and generalization in neural networks." Advances in neural information processing systems. 2018.

[*4] He, Kaiming, et al. "Delving deep into rectifiers: Surpassing human-level performance on imagenet classification." Proceedings of the IEEE international conference on computer vision. 2015.

[*5] Wu, Bichen, et al. "Fbnet: Hardware-aware efficient convnet design via differentiable neural architecture search." Proceedings of the IEEE Conference on Computer Vision and Pattern Recognition. 2019.

===================

Thanks for the rebuttal and the additional empirical results. In general, I really appreciate the "brave new idea" proposed by the authors, which could inspire new insights on neural architecture design.  However, my major concern still exists: the proposed metric seems to be sensitive to the choice of initialization functions. For example, uniform initialization [0, 1] seems not work at all but no further explanations, which may indicate the proposed method may work in a different way (e.g. taking advantage of some search space's bias).  So, I keep my original rating.

---

> ### Author Response · Authors · 2020-11-19
> **Author Response**
>
> We thank the reviewer for their time and comments. We are happy they found the direction interesting and will try to address their concerns here:
>
> Desire for more theoretical understanding: Complete formal theory for neural networks is a challenge at the best of times. However some understanding is still possible. The use of the covariance of derivatives captures an important aspect of the network, and in fact an aspect that is almost independent of the distribution of data inputs. For simplicity, let us ignore batchnorm units for the moment. The computational graph for a specific input (and equivalently for the derivative computation at that input) involves passing through many rectified linear units between the input and the output. Where those rectified linear units are zero, this “stops” computation at that point. Consider all the paths through this computational graph for two “nearby” points. Once the zeros are decided, the rest of the graph is linear. Hence can re-represent the derivative of the computation of the output with respect to the input as
>
> $\sum_{\text{path not zeroed out}} \prod_{\text{ij in path}} W_{ij}$
>
> For any one path, the only contribution to the correlation between those two points are paths where neither path passes through a zeroed rectified linear unit for that particular input. Hence when we compute the correlation we are computing a (slightly crude) statistical estimator for the shared paths at a point.
>
> Now for a network that has multiple shared paths between neighbouring points, a learning step for one point will strongly affect the other, as the parameters being updated are the same parameters. This works both ways - the update for the other data point affects the first. In this way it is hard for training to pull nearby points apart, and so networks with high correlations are potentially poor structures as they result in training difficulties. It seems the empirical evidence also reflects that.
>
> Of course, the above is speculative understanding, and we are keen not to introduce too much speculation in the paper, as it can be detrimental as ideas that are wrong can end up getting fixed within the community. We do think the use of the correlation of derivatives is a very valid measure; but equally we expect it to be surpassed by much better measures in the future. We also think that a good theoretical understanding, though very desirable, is not a prerequisite for an effective and valuable approach.
>
> The connection to the Neural Tangent Kernel is an interesting one: the idea that our correlation of gradient matrix $\Sigma$ could be used as the covariance matrix for a Gaussian Process was initially one of our considerations. However, in this paper our key goal was to demonstrate that a score of an untrained network could form the basis of a useable, simple NAS algorithm. We hope that the strength of the illustrations and empirical evidence we provided will be enough to motivate future theoretical excursions.
>
> Dependence on initialisation: It’s true that, since we take the score at initialisation, the specific initialisation function we use is critical. In hindsight, an investigation of our reliance on this should have been included in the paper. We have therefore added Appendix F to the paper, which considers 6 initialisation functions strategies. For the most part, the score is robust to initialisation function, with the exception of Uniform initialisation. In Figures 8 and 9 (Appendix C) we also show that aggregating the score over several re-initialisations does not reduce the efficacy of our score - this is to highlight that the dependence on specific initialisation is less important than the architectural properties themselves.
>
> To  address the notes
> - we change the output of the network so it is 1D for score computation.
> - We have added an exploration of how our score changes through training in Appendix H.

---

### Author Response · Authors · 2020-11-24
**Response Summary**

We’d like to thank all of the reviewers for their comments, and for providing concrete questions and detailed thoughts. The main shared query amongst the reviewers was a desire for further ablation testing: each reviewer had interesting questions which we have tried to answer by adding several appendices to the paper. We briefly summarise the questions here:

**How does performance change after aggregations over multiple images?**
To answer this question we have added several plots in Appendix C, where we show that the score is robust even to the use of random input data. This implies that the addition of multiple images (or even very diverse images) should not be more informative than a single randomly sampled subset from the training data. We also show the effect of aggregations over 20 images in Figure 9, where the impact on the effectiveness of our score is minimal. We believe we are capturing a property of the network structure, rather than something dataset-specific.

**How reliant is the method on the use of cutout augmentation?**
In Appendix I we show the effectiveness of the score when Gaussian noise and colour jitter are used for augmentation. The correlation between score and final test accuracy for cutout was 0.55, whereas for colour jitter it drops to 0.511 and for Gaussian noise it drops to 0.346. This illustrates that though cutout is the most effective augmentation scheme, the method is not completely reliant on it and it could be swapped out for other schemes in future.

**Is the method dependent on the choice of initialisation function?**
In Appendix F we consider a wide range of initialisation functions. We show that the score continues to be effective for most initialisation functions, except for Uniform weight initialisation between 0 and 1.

---

### Decision · Program_Chairs · 2021-01-07
**Final Decision**

**Decision:**

Reject

**Comment:**

This paper proposes an interesting new direction for low-cost NAS. However, the paper is not quite ready for acceptance in its current form. The main area of improvement is around the generalizability of the score presented, both empirically and (ideally) theoretically. The two main directions of generalizability that would be worth investigating are 1) different image datasets (see comments around Imagenet-16) 2) different/larger search spaces. Even simple search spaces consisting of a few architectural modification starting from standard architectures (e.g. resnets) would go a long way in convincing the community that the proposed method generalizes past NasBench.